# TRUST, BUT VERIFY: MODEL-BASED EXPLORATION IN SPARSE REWARD ENVIRONMENTS

## ABSTRACT

We propose *trust-but-verify* (TBV) mechanism, a new method which uses model uncertainty estimates to guide exploration. The mechanism augments graph search planning algorithms with the capacity to deal with learned model's imperfections. We identify certain type of frequent model errors, which we dub *false loops*, and which are particularly dangerous for graph search algorithms in discrete environments. These errors impose falsely pessimistic expectations and thus hinder exploration. We confirm this experimentally and show that TBV can effectively alleviate them. TBV combined with MCTS or Best First Search forms an effective model-based reinforcement learning solution, which is able to robustly solve sparse reward problems.

## 1 INTRODUCTION

Model-based approach to Reinforcement Learning (RL) brings a promise of data efficiency, and with it much greater generality. However, it is still largely an open question of how to make robust model-based RL algorithms. In most cases, the current solutions excel in low sample regime but underperform asymptotically, see Wang et al. (2019); Nagabandi et al. (2018a); Kaiser et al. (2020). The principal issues are the imperfections of the learned model and fragile planners not able to robustly deal with these imperfections, see (Sutton & Barto, 2018, Section 8.3), (François-Lavet et al., 2018, Section 6.2), (Wang et al., 2019, Section 4.5).

Model errors are unavoidable in any realistic RL scenario and thus need to be taken into account, particularly when planning is involved. They can be classified into two categories: optimistic and pessimistic, see (Sutton & Barto, 2018, Section 8.3). The former is rather benign or in some cases even beneficial, as it can boost exploration by sending the agent into falsely attractive areas. It has a self-correcting mechanism, since the newly collected data will improve the model. In contrast, when the model has a pessimistic view on a state, the agent might never have the incentive to visit it, and consequently, to make the appropriate adjustments. In this work, we propose *trust-but-verify* (TBV) mechanism, a new method to augment planners with capacity to prioritise visits in states for which the model is suspected to be pessimistic. The method is based on uncertainty estimates, is agnostic to the planner choice, and is rooted in the statistical hypothesis testing framework.

Taking advantage of the graph structure of the underlying problem might be beneficial during planning but also makes it more vulnerable to model errors. We argue that graph search planners might benefit most from utilising TBV. While TBV can correct for any type of pessimist errors, we explicitly identify certain type of model errors which frequently occur in discrete environments, which we dub *false loops*. In a nutshell, these are situations when the model falsely equates two states. Such errors are much similar to (Sutton & Barto, 2018, Example 8.3), in which the model erroneously imagine 'bouncing off' of a non-existing wall and thus failing to exploit a shortcut.

From a more general point of view, the problem is another incarnation of the exploration-exploitation dilemma. We highlight this by evaluating our method in sparse reward environments. In a planning context, exploration aims to collect data for improving the model, while exploitation aims to act optimally with respect to the current one. The problem might be solved by encouraging to revisit state-action pairs with an incentive of 'bonus rewards' based on the state-action visitation frequency. Such a solution, (Sutton & Barto, 2018, Dyna-Q+), is restricted to tabular cases. TBV can be though as its scalable version.

Our contributions are:

- TBV - a general risk-aware mechanism of dealing with pessimistic model errors, rooted in the statistical hypothesis learning framework.
- Conceptual sources and types of model errors accompanied with empirical evidence.
- Practical implementation of TBV with: classical Best First Search - a classical graph search algorithm family and Monte Carlo Tree Search - the state-of-the-art class of planners in many challenging domains.
- Empirical verification of TBV behaviour in two sparse rewards domains: ToyMontezumaRevenge Roderick et al. (2018) and the Tower of Hanoi puzzle.

The code of our work is available at: `https://github.com/ComradeMisha/TrustButVerify`.

## 2 RELATED WORK

There is a huge body of work concerning exploration. Fundamental results in this area come from the multi-arm bandits theory, including the celebrated UCB algorithm (Auer et al. (2002)) or Thompson sampling (Thompson (1933)), see also Lattimore & Szepesvári (2020) for a thorough treatment of the subject. There are multiple variants of UCB, some of which are relevant to this work, including UCB-V Audibert et al. (2007) and tree planning adaptations, such as UCT (Kocsis et al. (2006)) or PUCT (Silver et al. (2017; 2018)). Classical approach to exploration in the reinforcement learning setting, including the principle of optimism in the face of uncertainty and $\varepsilon$-greedy exploration, can be found in Sutton & Barto (2018). Exploration in the form of entropy-based regularization can be found in A3C (Mnih et al. (2016)) and SAC (Haarnoja et al. (2018)). Plappert et al. (2017) and Fortunato et al. (2017) introduce noise in the parameter space of policies, which leads to state-dependent exploration. There are multiple approaches relying on reward exploration bonuses, which take into account: prediction error (Stadie et al. (2015), Schmidhuber (2010) Pathak et al. (2019), Burda et al. (2018)), visit count (Bellemare et al. (2016), Ostrovski et al. (2017)), temporal distance (Machado et al. (2020)), classification (Fu et al. (2017)), or information gain (Sun et al. (2011), Houthooft et al. (2016)). An ensemble-based reinforcement learning counterpart of Thompson sampling can be found in Osband et al. (2016) and Osband et al. (2019). An exploration driven by experience-driven goal generation can be found in Andrychowicz et al. (2017). A set of exploration methods has been developed in an attempt to solve notoriously hard Montezuma's Revenge, see for example Ecoffet et al. (2019); Guo et al. (2019). In the context of model-based planning Lowrey et al. (2019) and Milos et al. (2019) use value function ensembles and uncertainty aware exploration bonuses (log-sum-exp and majority vote, respectively).

More recently, Pathak et al. (2017), Shyam et al. (2019), Henaff (2019) and Sekar et al. (2020) dealt with exploration and model learning. These works are similar in spirit to ours. They train exploration policies intending to reduce the model error (e.g. by maximizing the disagreement measure). Our work differs from the above in the following ways. First, we aim to use powerful graph search techniques as planners simultaneously to learning of the model. Our method addresses the fundamental issue, which is balancing intrinsic planner errors and the ones stemming from model imperfection. Second, we use the prediction error of the model ensemble to measure model uncertainty and apply a statistical hypothesis testing framework to switch between actions suggested by the model and actions leading to model improvement. Third, we use a discrete model and a discrete online planning regime.

When a learned model is unrolled during planning, errors typically accumulate dramatically. Numer of works adress this problem. Racanière et al. (2017) and Guez et al. (2018) are two approaches to learn the planning mechanism and make it robust to model errors. In a somewhat similar spirit, Eysenbach et al. (2019) treats the replay buffer as a non-parametric model forming a graph and uses ensembles of learned distances as a risk-aware mechanism to avoid certain types of model errors. Nagabandi et al. (2018b) successfully blends the strengths of model-based and model-free approaches. PlaNet (Hafner et al. (2019)) and Dreamer Hafner et al. (2020) train latent models, which are used for planning. Conceptually, a similar route was explored in MuZero (Schrittwieser et al. (2019)) and Universal Planning Networks (Srinivas et al. (2018)). Farquhar et al. (2017) and Oh et al.

(2017) investigate the possibility of creating neural network architectures inspired by the planning algorithms.

Many model-based reinforcement learning algorithms, including ours, follow the general framework laid out in Dyna, see Sutton (1991). Kaiser et al. (2020) uses a model to collect fictitious playouts and obtains impressive results for low data regime on Atari. In the continuous domains, a similar approach is adopted by Kurutach et al. (2018). Simultaneous training of a single policy on model ensemble is proposed to tackle model errors. Interestingly, this approach is reported to perform better than back-propagation through time. Clavera et al. (2018) indicates that polices trained on model ensembles might be over-conservative and proposed to use policy meta-learning. Another approach to dealing with model errors works by choosing the unroll horizon for a model. Janner et al. (2019), proposed short model-generated *branched rollouts* starting from data collected on "real environment". Perhaps surprisingly, they find that one-step rollouts provide competitive results. Similarly, short rollouts are used in Feinberg et al. (2018) and Buckman et al. (2018) to improve value estimates. Janner et al. (2019); Buckman et al. (2018) uses ensembles to reduce model bias. The former generates diverse data similarly to Chua et al. (2018), and the latter automatically adapt the planning horizon based on uncertainty estimates. Xiao et al. (2019) proposed another adaptive horizon mechanism based on measurement of model errors using principled Temporal Difference methods.

There is a huge body of work about planning; for classical results, the interested reader is referred to Cormen et al. (2009), Russell & Norvig (2003) and the references therein. Traditional heuristic algorithms such as A$^*$ (Hart et al. (1968)) or GBFS (Doran & Michie (1966)) are widely used in practice. In Agostinelli et al. (2019) the authors utilise the value-function to improve upon the A$^*$ algorithm and solve Rubik's cube. Similarly, Orseau et al. (2018) bases on the classical BFS to build a heuristic search mechanism with theoretical guarantees. The Monte Carlo Tree Search (MCTS) algorithm, which combines heuristic search with learning, led to breakthroughs in the field, see Browne et al. (2012) for an extensive survey. Famously, Silver et al. (2018) develop MCTS-based technique to master the ancient game of go. The POLO algorithm presented in Lowrey et al. (2019) proposes to enrich MPC planning with ensemble-based risk measures to augment exploration. This work is extended by NEEDLE (Milos et al. (2019)) to tree-based MCTS planners and by Lu et al. (2019) to successfully tackle life-long learning scenarios of environmental change. Hamrick et al. (2020) explores how to use better statistics collected while searching to calculate signal for value function training.

## 3 TBV FRAMEWORK

This work attempts to pave the way in harnessing the power of graph search methods to reinforce model-based reinforcement learning. Majority of the planners were designed to solve a well defined problem, which means that the user is expected to supply a *perfect model*. Consequently, when the user fails to do it, the planner can yield an unexpected, or outright wrong, result. Although in some environments, like games or simulators, a perfect model is available, in most real-world scenarios it is a luxury we cannot afford. Especially in the learning regimes, obtaining the perfect model requires exploration of the whole state space, which is usually unacceptably hard. Consequently, any real-world model-based RL system is unlikely to avoid dealing with model errors.

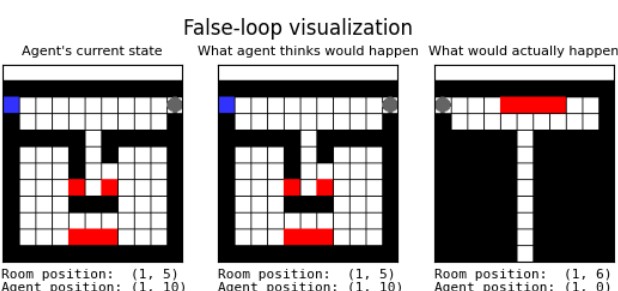

Figure 1: Agent (gray circle) is moving in a grid-world. It can move through the door (blue) to other rooms, if it enter a trap (red) the episode is terminated. (Genuine) one-step loop arise when agent attempts to walk into the wall. In the example, the model imagines that performing action "right" will not result in environment change (*false loop*). In fact choosing this action would result in moving to the next room.

We are thus led to two seemingly conflicting desiderata for the choice of a method: we would like to use planners which are robust to model errors, and planners which leverage the structure of the underlying problem, and hence are more susceptible to model errors. We advocate developing methods, which will facilitate adoption of search algorithms into setting with imperfect models. In this work we make a step towards implementing this plan. We identify and study certain type of model errors, which we dub *false loops*, and we propose *Trust, but verify* (TBV) method, which can successfully immunize planners with respect to said errors.

## 3.1 MODEL ERRORS

Sutton & Barto (2018, Section 8.3) conveniently categorize model errors into two categories: optimistic and pessimistic. A situation when a model predicts higher reward or better state transitions (optimistic error) is easier to handle since while the agent enters an erroneous region, it collects corrective data. More dangerously, predictions more pessimistic then reality may prevent the agent ever to visit certain regions.

The optimistic and pessimistic errors are often of the same nature. In this work, we concentrate on errors related to the graph structure of the problem. By definition, the edges in this graph correspond to transition induced by actions, and thus the errors are "incorrect edges". Perhaps the simplest prediction task (for a model) is to determine if an edge is *one-step loop* (i.e. if an action will change or not the current state). Another task is the prediction of absorbing states. In this work, we focused on the loop errors, as they turn out to be critical to performance (see Figure 1 for examples). In experiments, we also observed other types of errors (like "random teleports"); however they disappeared rather quickly during the training. It is possible that they would be more problematic in other environments, which we leave for further work. In Figure 5 we show numerical results that show rather frequent occurrence of *false loop*. This is a pessimistic error, which, unless corrected, e.g. by TBV, prevents exploration.

---

**Algorithm 1** Model-based training loop

```
# Initialize parameters of ensemble of models
# Initialize parameters planner specific networks
# Initialize buffer
repeat
    episode ← COLLECT_EPISODE
    buffer.ADD(episode)
    B ← buffer.BATCH
    Train ensembles of models
    Update planner specific networks
until convergence
function COLLECT_EPISODE
    s ← env.RESET
    episode ← []
    repeat
        a ← TBV_planner.CHOOSE_ACTION(s)          ▷ TBV-augmented graph planner
        s', r ← env.STEP(a)                        ▷ using models, see Algorithm 2
        episode.APPEND((s, a, r, s'))
        s ← s'
    until episode is done
    return episode
```

---

## 3.2 STATISTICAL HYPOTHESIS TESTING

Assume that there exists a function STATE_SCORE($s$) scoring each state by the model's uncertainty. Consider a state $s$ and denote the set of states reachable from $s$ in a specified budget of steps, by $D(s)$. In this paper, we will take $D(s)$ as a search graph expanded by the planner starting from $s$ and a standard deviation of model ensemble prediction error for STATE_SCORE, however other options are possible (e.g. states achievable by a behavior policy or information gain, respectively). The set $D(s)$, together with STATE_SCORE, imply the distribution of values of STATE_SCORE for the set

reachable form $s$. We are interested in determining whether the actions recommended by the planner at state $s$ leads to good exploration choices both in terms of the model (state-space exploration) and the value function (proper data for value function learning). To do this, we make use of statistical hypothesis testing. Namely, we form a null hypothesis, stating that the planner is exploring correctly, and an alternative hypothesis, making an opposite claim. If the STATE_SCORE($s$) takes extreme values, then it provides evidence against the null hypothesis. Hence, as a critical value for the test, we take the quantile of the STATE_STATE distribution on $D(s)$. This procedure has several interesting properties. It is invariant to the STATE_SCORE scale, provides an automatic method for threshold selection, and introduces only one task-agnostic hyperparameter. It can also mitigate the negative effect of propagating the errors in value function, which can deteriorate learning (see Lee et al. (2020) and Kumar et al. (2020)). Finally, since hypothesis testing is closely related to confidence intervals, our approach fits nicely with confidence bounds exploration methods.

### 3.3 TRUST, BUT VERIFY METHOD

*Trust, but verify* (TBV) is a method which can be integrated with graph search planners. The key idea of TBV is to prioritize visits in states for which the model is suspected to be pessimistic. In each state, the planner is asked for a recommended action. If the model uncertainty of the current state is high, we may suspect that the planner's results are misleading. We quantify it by an appropriate statistical test described in Section 3.2. This promotes exploration and thus reduces errors in the model. To measure uncertainty, we utilize ensembles, which empirically proved to be successful in that regard (see e.g. Osband et al. (2018)). The details follow the pseudo-code for TBV, listed in Algorithm 2. For completeness, we also present a typical model-based training loop in Algorithm 1.

As shown in Algorithm 2, TBV interacts with a planner, which uses an imperfect learned model. During its execution, the planner expands a search graph, $g_p$, in order to propose an action $a_p$. Each state-action pair in $g_p$ is given a score reflecting uncertainty assigned to it by the model, and which is quantified by a function DISAGREEMENT_MEASURE. Based on that, TBV decides whether to keep the planner's decision $a_p$ (trust the model) or override it (and explore to verify model predictions). More precisely, a state-action pair with the maximal score is considered if it reaches above the threshold given by $quantile\_score$. We found using quantiles instrumental in avoiding tuning thresholds which are problem-specific and change during the training.

Furthermore, $QR$ is relatively easy to tune (see Section 4) and is the only hyperparameter introduced by TBV. In the algorithm, we use additional randomization to prevent multiple revisits of the same state-action pair, while waiting for the model (and scores) update (see Section A.5 in Appendix for more detailed discussion). This could be alternatively realized by more principled approach, which we leave for future work.

Using TBV yields little computational overhead beyond the necessity of using ensembles (in efficient implementations STATE_SCORE and TRANSITION_SCORE calls can be inlined into $planner$.CHOOSE_ACTION). DISAGREEMENT_MEASURE depends on the state space representation; in our experiments, we found standard deviation computed on states to work well. Other methods, like Bayesian inference, could also be used.

### 3.4 BEST FIRST SEARCH

Best First Search (BestFS) is a family of search algorithms that build a graph by expanding the most promising unexpanded node. In the off-line setting, the search is run upon reaching the goal state. In order to apply BestFS in on-line framework, during each planning phase, we choose to extend the graph by a fixed amount of nodes, typically $10$. Afterwards, the recommended action is the first edge on the shortest path to the best node in the subgraph searched so far.

An important design decision is choosing heuristics for assigning nodes numerical values. This leads to many well-known algorithms (e.g., BFS, DFS, A$^*$, etc.). In our experiments, we concentrated on exploration and thus used `Disagreement_Measure` (standard deviation) between ensemble predictions. This choice proved to be effective in discrete problems with sparse rewards.

By design, BestFS ignores one step loops and thus is especially prone to *false loops* described in Section 1.

---

**Algorithm 2** TBV planner

---

**Require:**      $model$     ensemble of models used in Algorithm 1
                $planner$     planner that uses $model$
                $QR$     quantile rank $\in (0, 1)$
**Use:**       QUANTILE($l, q$)     computes the $q$-quantile of list $l$
                RANDOM()     random number from $\mathcal{U}(0, 1)$

  **function** CHOOSE_ACTION(state)
       $a_p \leftarrow planner$.CHOOSE_ACTION($state$)
       $g_p \leftarrow planner$.GET_GRAPH_OF_PLANNING()
       $scores \leftarrow []$
       **for** $state \in g_p$ **do**
           $scores$.APPEND(STATE_SCORE($state$))
       $quantile\_score \leftarrow$ QUANTILE($scores, QR$)
       $one\_step\_scores \leftarrow []$
       **for** $a \in actions$ **do**
           $one\_step\_scores$.APPEND(TRANSITION_SCORE($state, a$))
       **if** $\max(one\_step\_scores) > quantile\_score$ **and** RANDOM() $> 0.5$ **then**
          **return** $\arg\max_{action} one\_step\_score$
       **else**
          **return** $a_p$
  **function** TRANSITION_SCORE(state, action)
       $predictions \leftarrow []$
       **for** $network \in model$ **do**
           $next\_state, next\_reward \leftarrow network$.PREDICT_NEXT_STATE(state, action)
           $predictions$.APPEND(($next\_state, next\_reward$))
       **return** DISAGREEMENT_MEASURE($predictions$)
  **function** STATE_SCORE(state)
       **return** $\max_{action}$(TRANSITION_SCORE(state, action))

---

## 3.5 MONTE CARLO TREE SEARCH

Monte Carlo Tree Search (MCTS) is a well-established planning algorithm which was successfully applied to complex problems. Due to its simplicity and effectiveness, it has numerous extensions (see Browne et al. (2012) for a comprehensive review) including famous AlphaZero (Silver et al. (2018)).

A vanilla MCTS constructs a search tree in four stages: node selection, leaf expansion, rollout, and backpropagation (see Browne et al. (2012, Section 3.1)). In modern approaches, the rollout phase is often replaced by a neural network evaluation step. MCTS is often equipped with auxiliary mechanisms exploiting the graph structure of the problem. In our work, we use an MCTS implementation with four such mechanisms: hard loop-avoidance inside the search tree, soft loop-avoidance within to the whole episode, transposition tables and amortized value estimates, see Milos et al. (2019) for details.

Transposition tables (Childs et al., 2008; Gelly et al., 2012) enable sharing information between different tree nodes, which correspond to the same state (i.e. have been reached using distinct trajectories). Similar in spirit are mechanisms, which attempt to avoid visiting a node multiple times. These are akin to classical graph search methods in which a node it typically visited only once. This may take place on the level of the whole episode or within one planning phase (in the search tree). A "soft" way of implementing such mechanisms is to use "virtual loss", which assigns a temporary negative value when a graph node is first encountered. This encourages the exploration and tree building mechanism to avoid this node. Such solutions have been proposed in (Segal, 2010) to enhance parallelism and later used in (McAleer et al., 2019) to tackle Rubik's cube search. A hard version (equivalent to setting virtual loss to $-\infty$) strictly prohibits entering the same node twice. It (Milos et al., 2019) it is reported to be a significant efficiency boost for modest planning budgets on the classical Sokoban puzzle. Recent works (e.g. Milos et al. (2019), Hamrick et al. (2020)), propose to utilize "amortized value estimates" (which are calculated using the internal MCTS statistics) as targets for value function for training.

Our version uses a learned value function (similarly to AlphaZero) to evaluate new nodes. Similarly to Lowrey et al. (2019) and Milos et al. (2019) we use also value function ensemble to ensure exploration during planning.

## 4 EXPERIMENTS

The experiments below were designed to illustrate the aforementioned *false loops* model errors. They also demonstrate the effectiveness of TBV.

We have selected ToyMontezumaRevenge environment and the Tower of Hanoi puzzle (with 7 discs). These environments are challenging for RL as they have sparse rewards, and require at least a few hundred steps to reach the solution. At the same time, they are solvable through an exhaustive search using graph search algorithms with access to a perfect model. Since there is a gap in the performance of these two categories of methods, the environments present a good testing grounds for studying the unfavorable influence of model errors on graph planners.

As for performance metrics, we use minimal distance to the goal. It is measured in steps and treated as a function of total environment steps

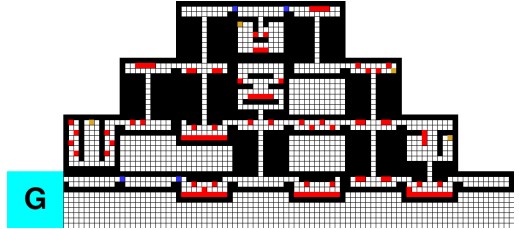

Figure 2: Map of ToyMontezumaRevenge state space. The only reward is given after reaching the goal room marked with $G$. To obtain it, agent needs to gather several keys (marked in yellow) and go through doors (marked in blue). Stepping on a trap (marked in red) results in episode termination.

used in training. We apply this measure to two planners: BestFS and MCTS (each in three versions: with TBV, with $\varepsilon$-greedy and without additional exploration mechanism). As an additional baseline, we also use RND (see Burda et al. (2018)), a strong model-free exploration algorithm.

In all experiments, we found that using TBV mechanism significantly outperform other baselines.

### 4.1 TOYMONTEZUMAREVENGE

ToyMontezumaRevenge is a navigation, maze-like, environment introduced by Roderick et al. (2018).[1] It is a testing ground for long-horizon planning and exploration (see Figure 2). It has a greatly simplified visual layer when compared to the original Montezuma's Revenge Atari game, but it retains much of its exploration difficulty. In our experiments, we consider the biggest map containing 24 rooms and sparse rewards: the agent gets a reward 1 only if it reaches the treasure room; otherwise the episode is terminated after 600 steps. Observation is represented as a tuple containing current room location, agent position within the room, and status of all keys and doors on the board, see Figure 2 for the full map of the state space.

For both planners, MCTS and BestFS, using $\varepsilon$-greedy improves over baseline version, but not as much as TBV (see Figure 3).

It turns out that traps (see the caption below Figure 1) pose a challenge for BestFS algorithm. In order to mitigate this issue, we modified the TBV mechanism for BestFS in this environment to explicitly avoid traps (performing actions resulting in episode termination without positive reward, according to the learned model). For a fair comparison, we also added the same mechanism to $\varepsilon$-greedy variant: when sampling from the action space, only the actions not leading to traps were considered. Similar improvements were unnecessary for MCTS.

For BestFS, 4 out of 10 experiments were unable to leave the first room. This stemmed from the fact that the agent could not make progress: it kept getting stuck in the early stages of exploration due to the inability to pass through *false loops*. This is visualized in Figure 5, where it is shown that TBV encourages the agent to leave the first room when it has a chance to do so but is being blocked by the model error.

---

[1]We use the code from `https://github.com/chrisgrimm/deep_abstract_q_network`

Interestingly, it was quite hard to tune RND on this domain (the agent struggled to find the second key and explored at most 14 out of 24 rooms). This could be related to the fact that RND needs a lot of data (originally RND was trained with billions of transitions), whereas our experiments are conducted in low data regime. Hyperparameters choice for RND can be found in Appendix (see Table 1, Table 2 and Table 3).

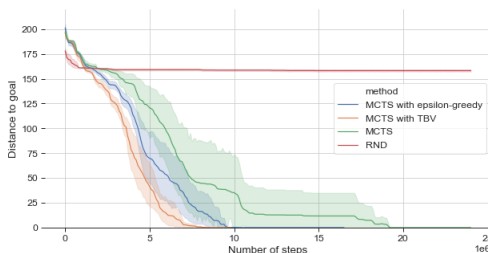 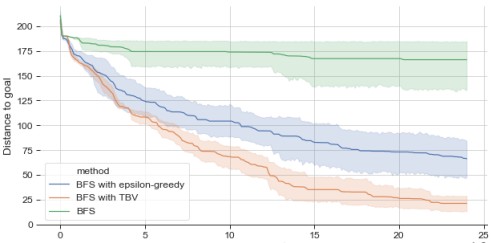

Figure 3: ToyMontezumaRevenge, comparison between planners augmented with *TBV*, $\varepsilon$-greedy, and no top-level exploration mechanism. Results are averaged over 10 random seeds, shaded areas show 95% confidence intervals.

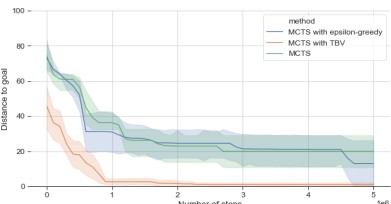 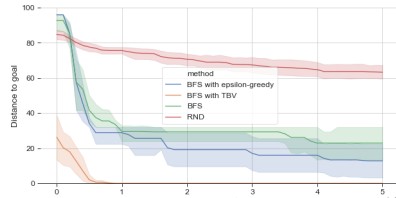

Figure 4: Tower of Hanoi, both MCTS and BestFS quickly find the solution if augmented with *TBV*. Other methods struggle to find solution. Results are averaged over 10 random seeds, shaded areas show 95% confidence intervals.

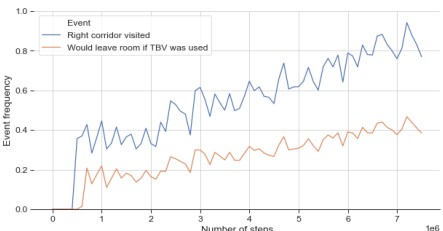 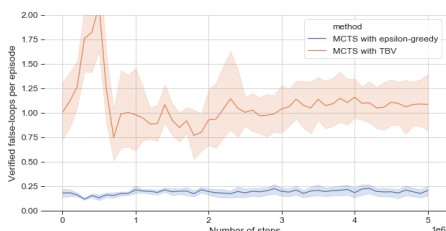

Figure 5: Numerical measurement of influence of TBV on BestFS in the presence of a *false loop* in ToyMR. The upper (blue) line presents the frequency of reaching the open doors in the first room for the agent without TBV. This agent never enters the door due to a *false loop* error. The lower (orange) line present the frequency the agent would leave the room if TBV was used.

Figure 6: *False loop* verification in the Tower of Hanoi. We consider two agents: MCTS with TBV (orange) and $\varepsilon$-greedy (blue) exploration (presented also on Figure 4). The plots shows how often agent performed action which falsely seemed to be a loop in the state-action graph. TBV does much better job in verifying such transitions.

## 4.2 TOWER OF HANOI

The Tower of Hanoi is a classical puzzle consisting of 3 pegs and $n$ disks (see Figure 7). The objective of the game is to move the entire stack of disks from the starting peg to the goal peg. The rules are that only one disk can be moved at a time and it is not allowed to put a larger disk on top of a smaller one. It can be shown that an optimal solution requires $2^n - 1$ moves (see e.g. Pierrot et al. (2019)). This makes it a challenging combinatorial problem and an interesting domain for planning

methods. Model-free algorithms struggle to deal with larger instances (Troussard et al. (2020) used $n = 4$, and Edwards et al. (2018) used $n = 3$). In this paper, we use a considerably harder version, with $n = 7$. The observation passed to the agent is represented as a binary vector of length $3 \times n$. Each consecutive triplets of bits encode the location (peg) of a given disk. This representation makes the task more difficult for the environment model as we do not pass disk sizes. This prevents easy generalization of the dynamics of the environment from partial data, and it makes it challenging to learn the rule that "the smaller disk can always be placed on larger". Instead, the agent needs to learn the relation between sizes for each pair of disks separately, as the exploration progresses.

As can be seen in Figure 4, TBV method significantly improves both MCTS and BestFS planners in this domain. This is due to the nature of the Tower of Hanoi, which in the early stages of training creates an illusion that the larger disks stays in the same position. It may result in the false belief of the model that the larger disks cannot be moved, hence causing the *false loop* errors. We verified that TBV handles such errors effectively, see Figure 6. We found that RND continued to explore the environment when left for longer training (up to 25M transitions, see Figure 10 in the Appendix.). Its progress is, however, very slow when compared to other methods. This is in line with our observation from the previous section that RND is designed to work on larger scale experiments.

### 4.3 QUANTILE RANK SENSITIVITY

By design, TBV it quite easy to tune. Its only hyperparameter is Quantile Rank (QR). We found out that values in the range $0.8 - 0.95$ worked well across most choices of environments and planners. Excessively large value of QR (e.g. 0.99) causes TBV to accept planner's actions too often, slowing down exploration. Performance of a BestFS agent with *TBV* across multiple values of QR is presented in Appendix in Figure 8.

Figure 7: Tower of Hanoi puzzle. For the first time, the agent finds himself in a position to move the fourth disk but mistakenly believes that it is not possible, resulting in a false-loop.

## 5 CONCLUSIONS

This work concentrates on an important open research problem, concerning the design of planners, which are resistant to model errors. Our study shows that model imperfections, especially *false loops*, can significantly hinder exploration. New algorithm presented in this paper, TBV, alleviates them by repeatedly verifying model predictions. TBV builds upon a statistical hypothesis testing framework and uses disagreement measure distribution to form an appropriate test.

Based on experiments in two challenging domains, we verify the design to be successful. Precisely, we show experimentally, that augmenting a planner with TBV, significantly improves agents exploration. At the same time we empirically confirm that TBV actively encourages the agent to pass through *false loops*.

Finally, TBV is a robust algorithm: it can be combined with different model-based planners and is easy to tune.

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

## A   APPENDIX

### A.1   ALGORITHMS

We tested the combination of TBV with two planning algorithms: Monte Carlo Tree Search (equipped with some auxiliary mechanisms) and Best First Search. Here we present the pseudo-code for both algorithms and provide some additional explanation.

#### A.1.1   BESTFS

In Algorithm 3 we present the pseudocode for BestFS algorithm used in out experiments. In Data Structures 4 and 5 we present two additional data structures used by our implementation of BesFS.

---

**Algorithm 3** BestFS planner

---

**Require:**    $model$    learned model used in Algorithm 1
                $V$    value function
                $C$    number of nodes expansions per step
                $\gamma$    discount factor
**Use:**          $F$    priority queue of unexpanded nodes (fringe)
                $G$    graph of all seen nodes

**function** RESET
    $F \leftarrow \emptyset$
    $G \leftarrow \emptyset$
**function** CHOOSE_ACTION(state)
    $G$.ADD($state$)
    **if** $state \notin G$ **then**
        $state.n\_visits \leftarrow 1$
    **else**
        $state.n\_visits \leftarrow state.n\_visits + 1$
    $reachable \leftarrow$ FIND_REACHABLE_NODES($G, state$)    ▷ find all nodes reachable from state
    EXPAND_GRAPH($reachable$)
    $best\_node \leftarrow G$.FIND BEST NODE($reachable$)    ▷ see Algorithm 5
    $path \leftarrow$ FIND_PATH_TO_NODE($best\_node$)
    **return** $path.first\_action$
**function** EXPAND_GRAPH(reachable)
    **for** $1 \ldots C$ **do**
        $n \leftarrow F$.POP_BEST_NODE($reachable$)    ▷ see Algorithm 4
        EXPAND_GRAPH_NODE($n$)
**function** EXPAND_GRAPH_NODE(n, reachable)
    $children \leftarrow [\,]$
    **for** $a \in \mathcal{A}$ **do**
        $model$.LOAD_STATE($n.state$)
        $new\_state \leftarrow model$.STEP($a$)
        $new\_state.uncertainity \leftarrow \max_{action} model.uncertainity($new_state, action$)$
        **if** $new\_state \notin G$ **then**
            $value \leftarrow V$.EVALUATE($new\_state$)
            $new\_state$.SET_VALUE()
            $F$.ADD_NEW_NODE($new\_state$)
            $G$.ADD_NEW_NODE($new\_state$)
            $reachable$.ADD($new\_state$)
    $F$.REMOVE($n$)    ▷ fringe contains only unexpanded nodes
**function** GET_GRAPH_OF_PLANNING()
    **return** $G$

---

---

**Data structure 4** Fringe (priority queue)

**Require:** $S_F \triangleright$ set of nodes
1: **function** ADD_NODE(n)
2: $\quad S_F$.ADD($n$)
3: **function** REMOVE(n)
4: $\quad S_F$.REMOVE($n$)
5: **function** POP_BEST_NODE(reachable)
6: $\quad nodes \leftarrow S_F$.INTERSECTION($reachable$)
7: $\quad$ **return** $\arg\max_{n \in nodes} n.uncertainity$

---

**Data structure 5** BestFS graph

**Require:** $S_G \triangleright$ set of nodes
1: **function** ADD_NODE(n)
2: $\quad S_G$.ADD($n$)
3: **function** LEXICOGRAPHICAL_MAX(nodes) $\quad \triangleright$ returns max according to keys: (solved, -n_visits, uncertainity)
4: **function** FIND_BEST_NODE(reachable, state)
5: $\quad nodes \leftarrow S$.INTERSECTION($reachable$)
6: $\quad$ **return** LEXICOGRAPHICAL_MAX($nodes$)

---

### A.1.2 MCTS WITH AVOID HISTORY COEFFICIENT DEAD ENDS AND AVOID SELF LOOPS

In Algorithm 6 we present the general structure of our MCTS planner. In Algorithms 7, 10, 8 and 9 we present our version of key functions in MCTS. In Algorithm 11 we provide pseudo code for ad auxiliary dead end mechanism used in this paper. Finally, in Data structure 12 we present the functionality of graph node used in this implementation.

---

**Algorithm 6** MCTS planner

| **Require:** | $model$ | learned model used in Algorithm 1 |
| | $V$ | value function ensemble (assigns a vector of value estimates to state) |
| | $C$ | number of MCTS passes |
| | $\kappa$ | score factor for value uncertainty |
| | $\gamma$ | discount factor |
| | $\gamma_{dead}$ | dead end value |
| | $\gamma_{ah}$ | avoid history coefficient |
| **Use:** | $T$ | planning graph |
| | $S_{visited}$ | set of states visited by the agent |
| | $S_{seen}$ | set of states seen in one MCTS pass |

  **function** RESET
    $T \leftarrow \emptyset$
  **function** CHOOSE_ACTION(state)
    $root \leftarrow state$
    **for** $1 \dots$ C **do**
      MCTS_PASS($state$)
    **return** $\arg\max_{a \in \mathcal{A}}(n.child(a).value)$
  **function** MCTS_PASS(root)
    $S_{seen} \leftarrow \emptyset$                                           $\triangleright$ used in Algorithm 11
    $path, leaf \leftarrow$ TRAVERSAL($root$)
    $value \leftarrow$ EXPAND_LEAF($leaf, model$)
    BACKPROPAGATE($value, path$)
  **function** GET_GRAPH_OF_PLANNING() **return** $T$

---

**Algorithm 7** traversal()

**Input:** root

1: n ← root
2: path ← ∅
3: **while** n is not a leaf **do**
4:     $a$ ← select_child($n$)
5:     **if** $a$ is None **then** ▷ dead end, terminal or leaf
6:         **break**
7:     $path$.APPEND($(n, a)$)
8:     $n$ ← $n.child(a)$
9: **return** $path, n$                    ▷ n ∉ path

---

**Algorithm 8** expand_leaf()

**Require:** $\gamma_{dead}, V, model$
**Input:** leaf        ▷ MCTS tree node without children

1: **if** $leaf$ is terminal **then**
2:     UPDATE($leaf, 0.$)
3:     **return** 0.
4: **else if** IS_DEAD_END($leaf$) **then**
5:     UPDATE($leaf, \gamma_{dead}$)
6:     **return** $\gamma_{dead}$
7: **else**
8:     **for** $a \in \mathcal{A}$ **do**
9:         $new\_node$ ← CREATE_NODE()
10:         $model$.LOAD_STATE($leaf.state$)
11:         $new\_state$ ← $model$.STEP($a$)
12:         $new\_node.state$ ← $new\_state$
13:         **if** $new\_state$ not yet visited **then**
14:             $new\_state.value$ ← $V(new\_state)$
15:         $leaf.child(a)$ ← $new\_node$
16: **return** $leaf.value$

---

**Algorithm 9** backpropagate()

**Require:** $\gamma$
**Input:** $v, path$

1: **for** $(n, a)$ in reversed($path$) **do**
2:     $v$ ← $n.reward + \gamma v$
3:     $n$.UPDATE($v$)

---

**Algorithm 10** select_child()

**Input:** n

1: **if** $n$ not expanded or terminal **then**
2:     **return** None
3: **if** IS_DEAD_END($n$) **then**
4:     **return** None
5: **else**
6:     $\mathcal{A}_{allowed}$ ← ALLOWED($n$)
7:     **return**
      $\arg \max_{a \in \mathcal{A}_{allowed}} n.child(a).$EVAL()

---

**Algorithm 11** Dead end detection

**Require:** $S_{seen}$
**Input:** $v, path$

1: **function** ALLOWED(n)
2:     $\mathcal{A}_{allowed}$ ← ∅
3:     **for** $a \in \mathcal{A}$ **do**
4:         **if** $n.child(a) \notin S_{seen}$ **then**
5:             $\mathcal{A}_{allowed}$.ADD($a$)
6:     **return** $\mathcal{A}_{allowed}$
7: **function** IS_DEAD_END(n)
8:     $\mathcal{A}_{allowed}$ ← ALLOWED($n$)
9:     **if** $\mathcal{A}_{allowed} \neq \emptyset$ **then**
10:         **return** False
11:     **else**
12:         **return** True

---

**Data structure 12** node

**Attributes:**   $count$   MCTS node counter
                  $value$   value ensemble estimate
                  $visits$  number of agents visits
                  $child$   list of children nodes

1: **function** UPDATE(value)
2:     $self.value$ ← $self.value + value$
3:     $self.count$ ← $self.count + 1$
4: **function** ADD_VISIT()
5:     $self.visits$ ← $self.visits + 1$
6: **function** EVAL()
7:     $s$ ← SCORE($self.value, self.count$)
8:     $std$ ← STD($self.value$) ▷ uncertainty
      of value ensemble
9:     **return** $score + \gamma_{ah} self.visits + \kappa std$

## A.2 TRAINING SETUP

Code for all our experiments can be accessed at `https://github.com/ComradeMisha/TrustButVerify`.

Our experiments adhere to the general model-based training loop logic, described in Algorithm 1. We use a distributed system with 32 workers solving distinct episodes, where data gathered across a batch of workers is collected in two common experience buffers: the replay buffer for trainable model of size 50000 and the replay buffer for value function of size 30000. Before the solving of actual episode, we collect 1000 random trajectories, that we use for initial training of the model of

environment. This initial training has critical importance for the performance of our agents. Episodes are limited to: 600 steps for ToyMontezumaRevenge and 1000 steps for the Tower of Hanoi.

To update value network for MCTS we follow approach of Milos et al. (2019) (similar to Hamrick et al. (2020)) for BestFS experiments we simply calculate future discounted returns for each transition encountered by agent.

### A.3 NETWORK ARCHITECTURES FOR MODEL AND VALUE

#### A.3.1 SINGLE NETWORKS

In every experiments we use two neural network architectures: one to estimate the value function of a given state and the other to predict the outcome of taking a step in the environment. In ToyMR we represent the state as a vector of length 17 and in the Tower of Hanoi as a vector of length $3 \times$ number of discs (it is due to one-hot encoding of the peg number for each disc).

In both environments, for value estimation we use an MLP (multilayer perceptron) architecture with two hidden layers of 50 neurons and ReLu non-linearity. For model, in both environments, we use an MLP with four hidden layers of 250 neurons and ReLU activation functions.

The model network has three outputs: the difference between next and current observations (delta target), reward and the episode termination flag (0 or 1).

#### A.3.2 ENSEMBLE

For both value estimation and model we use ensemble of networks with architectures described in Section A.3.1. The usage of ensembles is however different for value network and model network. Ensemble of model network is used as described in Algorithm 2. In case of value estimation, the final result of value is an average of predictions across ensemble. The standard deviation of this predictions multiplied by $\kappa$ is used by MCTS as an auxiliary score added to each vertex in the search tree.

In Algorithm 13 we present the procedure of transforming the output of ensemble of model networks to a valid prediction of environment signal. In our experiments, it turned out to be beneficial to train the model to predict change between observations rather then the ready observations. The transformation performs the following step: collects predictions of all networks in ensemble, then averages next predicted change of observations, predicted rewards and predicted end of episode flags. Averaged observation change is added to current state, then clipped to the range of possible values (some coordinates of the state vector are binary variables), rounded to integers and returned as the next predicted state. Reward and end of episode flag are predicted to be true if their average value across networks in ensemble is larger than $0.5$.

To stabilize performance of value and model ensembles we used additional masks mechanism. We create some number of networks (given by ensemble size parameter), that are trained with the same data buffer, but for prediction each worker uses only a random subset of given size (given by number of masks parameter).

### A.4 QUANTILE RANK VALUE ANALYSIS

Figure 8 presents performance of BestFS Agent in the Tower of Hanoi environment for different values of Quantile Rank. We observed the the best performance was for values between 0.8 and 0.95, and when the parameter was within the appropriate range of values, the performance was not very sensitive to the changes of quantile rank parameter. We observed similar behaviour in ToyMR experiments - the best performance was for quantile rank between 0.9 and 0.95 but it was hard to narrow down this interval, due to aforementioned little sensitivity.

**Algorithm 13** Transforming model ensemble predictions

---

**function** PREDICT_STEP(state, action)
    $next\_observations \leftarrow []$
    $rewards \leftarrow []$
    $is\_done\_flags \leftarrow []$
    **for** $network \in ensemble$ **do**:
        $(next\_obs\_delta, reward, is\_done) \leftarrow network.\text{PREDICT}(state, action)$
        $next\_obs \leftarrow state + next\_obs\_delta$
        $next\_observations.\text{APPEND}(next\_obs)$
        $rewards.\text{APPEND}(reward)$
        $is\_done\_flags.\text{APPEND}(is\_done)$
    $predicted\_reward \leftarrow \text{AVERAGE}(next\_reward)$
    **if** $predicted\_reward > 0.5$ **then**
        $predicted\_reward \leftarrow 1$
    **else**
        $predicted\_reward \leftarrow 0$
    $predicted\_is\_done \leftarrow \text{AVERAGE}(is\_done\_flags)$
    **if** $predicted\_is\_done > 0.5$ **then**
        $predicted\_is\_done \leftarrow True$
    **else**
        $predicted\_is\_done \leftarrow False$
    $predicted\_next\_obs \leftarrow \text{AVERAGE}(next\_observations)$
    $predicted\_next\_obs \leftarrow \text{CLIP}(predicted\_next\_obs)$
    $predicted\_next\_obs \leftarrow \text{ROUND}(predicted\_next\_obs)$
    **return**$(predicted\_next\_obs, predicted\_reward \ predicted\_is\_done)$

---

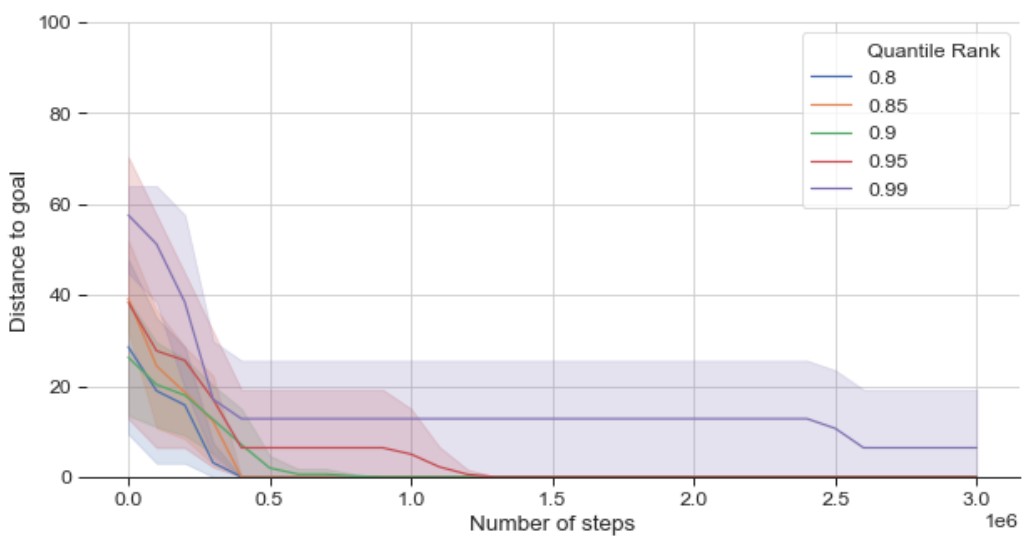

Figure 8: Performance of *TBV* BestFS on the Tower of Hanoi domain with different values of Qantile Rank

## A.5   RANDOM TBV REJECTION ANALYSIS

In our preliminary experiments we have seen that without random rejection of TBV mechanism the agents often tends to get stuck in cycles with high ensemble disagreement and is unable to leave such cycle until the end of episode (since model is not updated in during the episode). In Figure 9 we present the performance in Tower of Hanoi for different values of frequency of random TBV

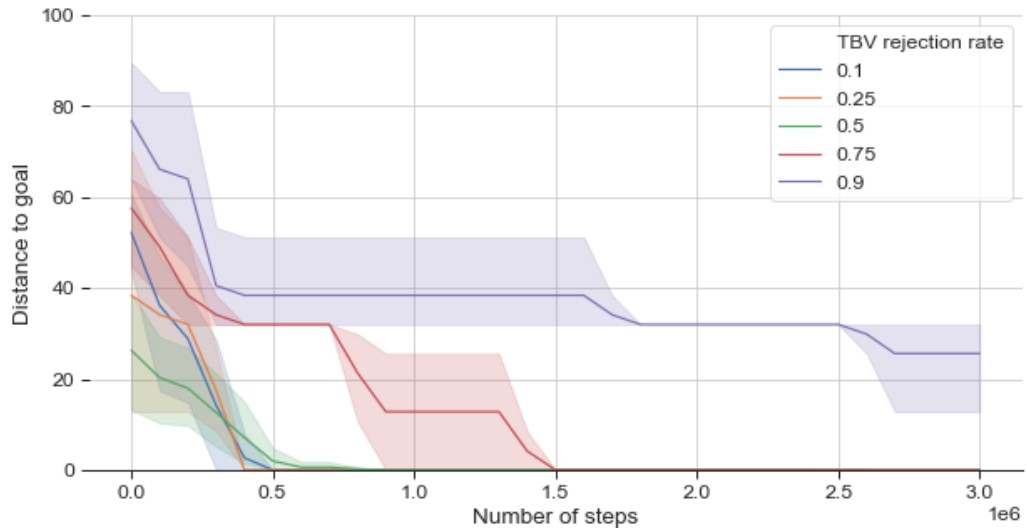

Figure 9: Performance of *TBV* BestFS on the Tower of Hanoi domain with different frequencies of random rejection of TBV

rejection (while the rest of parameters is same as in our best experiments). It can be seen, that if the frequency of TBV override (i.e. frequency with which we allow TBV to change planners action) is between 0.5 and 0.9 we obtained best results. Also, TBV is not very sensitive to this parameter as long it is in an appropriate range of values.

## A.6   HYPER-PARAMETERS

In tables 1 and 2 we present hyper-parameters used in our experiments.

| Parameter | Toy MR BestFS | Toy MR MCTS | Tower of Hanoi BestFS | Tower of Hanoi MCTS |
|---|---|---|---|---|
| Number of planner passes $C$ | 10 | 10 | 10 | 10 |
| Discounting factor $\gamma$ | 0.99 | 0.99 | 0.99 | 0.99 |
| $\varepsilon$ greedy exploration for baslines | 0.001 | 0.02 | 0.001 | 0.02 |
| score factor for value uncertainty $\kappa$ | - | 3 | - | 3 |
| Dead end value | - | $-0.2$ | - | $-0.2$ |
| Avoid history coefficient | - | $-0.2$ | - | $-0.2$ |
| Value ensemble size [2] | 20 | 20 | 20 | 20 |
| Value ensemble mask size [2] | 10 | 10 | 10 | 10 |
| Model ensemble size [2] | 8 | 8 | 8 | 8 |
| Model ensemble mask size [2] | 4 | 4 | 4 | 4 |
| Optimizer | RMSprop | RMSprop | RMSprop | RMSprop |
| Learning rate | 2.5e$-$4 | 2.5e$-$4 | 2.5e$-$4 | 2.5e$-$4 |
| Batch size for value training | 32 | 32 | 32 | 32 |
| Batch size for model training | 1024 | 1024 | 1024 | 1024 |

[2] See Section A.3.2 for explanation of these parameters.

Table 1: Hyper-parameters values used in our experiments.

For MCTS we took hyperparameters from Milos et al. (2019). As they used setup without learned model we needed to tune model architecture and model training parameters on ToyMontezumaRe-

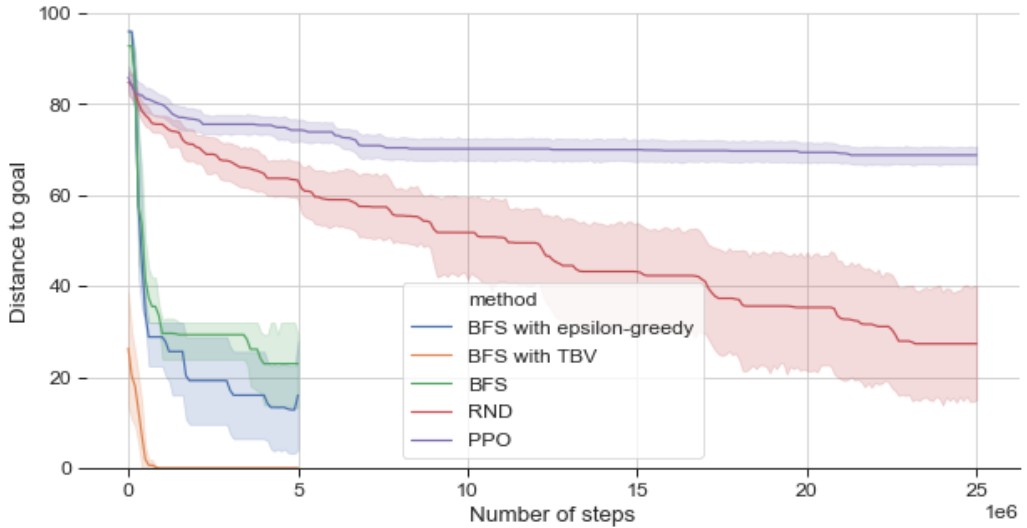

Figure 10: Performance of RND and PPO on Tower of Hanoi with longer training.

venge domain. Then, we separately tuned epsilon and Quantile Rank for each combination of domain/planner.

## A.7 RND AND PPO

RND continued to further explore state space for Hanoi (figure 10), for comparison we also show performance of PPO (the same parameters as RND but without intrinsic reward).

| Parameter | ToyMR | Tower of Hanoi |
|---|---|---|
| Rollout length | 128 | 128 |
| Maximal length of an episode | 600 | 1000 |
| Total number of rollouts per environment | 6200 | 6200 |
| Number of minibatches | 4 | 4 |
| Number of optimization epochs | 16 | 4 |
| Coefficient of extrinsic reward | 1 | 1 |
| Coefficient of intrinsic reward | 100 | 10 |
| Number of parallel environments | 32 | 32 |
| Learning rate | 0.001 | 0.001 |
| Optimization algorithm | Adam | Adam |
| $\lambda$ | 0.95 | 0.95 |
| Entropy coefficient | 0.001 | 0.001 |
| Proportion of experience used for training predictor | 1.0 | 1.0 |
| $\gamma_E$ | 0.999 | 0.999 |
| $\gamma_I$ | 0.99 | 0.99 |
| Clip range | [0.9, 1.1] | [0.9, 1.1] |
| Policy architecture | FCN | FCN |

Table 2: Default hyper-parameters for PPO and RND algorithms for ToyMR and Tower of Hanoi experiments.

| Parameter | Value |
|---|---|
| Number of optimization epochs | $[1, 4, 16]$ |
| Coefficient of intrinsic reward | $[1, 3, 10, 30, 100, 300]$ |
| Learning rate | $[5 \cdot 10^{-2}, 10^{-2}, 5 \cdot 10^{-3}, 10^{-3}, 5 \cdot 10^{-4}, 10^{-4}]$ |
| $\lambda$ | $[0.95, 0.99]$ |
| Proportion of experience used for training predictor | $[0.25, 1.0]$ |
| $\gamma_E$ | $[0.999, 0.9999]$ |
| $\gamma_I$ | $[0.99, 0.999]$ |
| Policy architecture layers number | $[2, 3, 4]$ |
| Policy architecture layers width | $[64, 128, 256]$ |
| Random target and prediction networks last layer width | $[64, 128]$ |

Table 3: Hyper-parameters for RND algorithm checked during tuning process of ToyMontezumaRevenge.

