# OpenReview forum: "Trust, but verify: model-based exploration in sparse reward environments"
_ICLR.cc/2021/Conference — Reject_

### Official Review · AnonReviewer3 · 2020-10-28
**Improperly positioned, minor technical contribution, unclearly written**

**Rating:** 2
**Confidence:** 4

**Review:**

This paper proposes an approach for encouraging exploration when planning over learned models of discrete reinforcement learning environment. The proposed method involves using an uncertainty-aware model (e.g., an ensemble of neural networks) to predict state-action transitions, together with a graph-based planner operating on this model. The key idea is to replace (with some probability) the planner's action with the action leading to the highest uncertainty in model prediction. The paper evaluates the proposed technique using two standard search planners (MCTS and BFS).

Unfortunately, I think the significance and technical contribution of this work is minimal, an issue that mostly likely starts from a deficient literature review. What the authors refer to as Trust-But-Verify, it's just an ad-hoc instance of the well-known principle of *optimism in the face of uncertainty*, which underlies classic bandit and RL algorithms such as UCB1[1], UCT[2], Thompson Sampling [3, 4]. In the model-free setting, this idea has lead to numerous recent algorithms, many of which also use ensembles for uncertainty quantification [5-8]. In the model-based setting there are also some precedents of work using similar ideas [9-11]. There is also a large body of work treating the problem from the point of view of Bayesian RL (see [12] for a survey).  It is a bad sign that none of this body of previous work was discussed in the paper, which I would argue was the more relevant literature upon which the paper had to be positioned.

This could conceivably be excused if the paper technical and experimental contribution was impressive enough, but this is not the case.  In contrast to the literature outlined above (where proposed exploration strategies typically follow from rigorous statistical analysis), this work presents the proposed method as a heuristic rule, providing no insight as to why one should expect the approach to work well in general. Moreover, the experiments are done in relatively simple domain, and compared against simple baselines. Some of the baseline choices do not seem appropriate either. For example, why use $\epsilon$-greedy for exploration, instead of more robust strategies using upper confidence bounds?

Finally, the writing on the paper can be improved in many places. For example, the paper refers to using the graph structure of the underlying problem, but what this graph structure refers to is not properly defined anywhere in the paper. I imagine it refers to the graph wherein edges represent non-zero probability transitions between states, but this is not clear from the text. Additionally, some paragraphs add little in terms of content. For example, the first paragraph of Section 3 is devoted to describe the basic problem that all model-based RL methods are trying to solve; this issue is ubiquitous so there is no need for a full example and so much text to describe this. Other sentences are unclear, such as "The optimistic and pessimistic errors are often of the same nature", which I am not sure what is referring to . Additionally, I didn't see a description of the learned model used in the experiments, which is not an obvious choice, since the environment is discrete.

Overall, to end on a somewhat constructive note, I think the problem the authors are trying to solve is interesting and the proposed approach is based on the right intuitions. However, this work is still too immature for publication. I suggest to the authors to position their work properly with regards to the relevant literature, refine their technical contribution accordingly, and compare with more appropriate baselines.

----------------------------------------------------------------------
[1] Auer, Peter, Nicolo Cesa-Bianchi, and Paul Fischer. "Finite-time analysis of the multiarmed bandit problem." Machine learning 47.2-3 (2002): 235-256.

[2] Kocsis, Levente, and Csaba Szepesvári. "Bandit based monte-carlo planning." European conference on machine learning. Springer, Berlin, Heidelberg, 2006.

[3] Thompson, William R. "On the likelihood that one unknown probability exceeds another in view of the evidence of two samples." Biometrika 25.3/4 (1933): 285-294.

[4] Russo, Daniel, et al. "A tutorial on thompson sampling." arXiv preprint arXiv:1707.02038 (2017).

[5] Bellemare, M., Srinivasan, S., Ostrovski, G., Schaul, T., Saxton, D., & Munos, R. (2016). Unifying count-based exploration and intrinsic motivation. In Advances in neural information processing systems (pp. 1471-1479).

[6] Ostrovski, G., Bellemare, M. G., Oord, A., & Munos, R. (2017, July). Count-Based Exploration with Neural Density Models. In International Conference on Machine Learning (pp. 2721-2730).

[7] Osband, I., Blundell, C., Pritzel, A., & Van Roy, B. (2016). Deep exploration via bootstrapped DQN. In Advances in neural information processing systems (pp. 4026-4034).

[8] Fortunato, M., Azar, M. G., Piot, B., Menick, J., Osband, I., Graves, A., ... & Blundell, C. (2017). Noisy networks for exploration. arXiv preprint arXiv:1706.10295.

[9] Sanner, S., Goetschalckx, R., Driessens, K., & Shani, G. (2009). Bayesian real-time dynamic programming. In Proceedings of the 21st International Joint Conference on Artificial Intelligence (IJCAI-09) (pp. 1784-1789). IJCAI-INT JOINT CONF ARTIF INTELL.

[10] Pathak, Deepak, Dhiraj Gandhi, and Abhinav Gupta. "Self-supervised exploration via disagreement." arXiv preprint arXiv:1906.04161 (2019).

[11] Shyam, Pranav, Wojciech Jaśkowski, and Faustino Gomez. "Model-based active exploration." International Conference on Machine Learning. 2019.

[12] Ghavamzadeh, M., Mannor, S., Pineau, J., & Tamar, A. (2016). Bayesian reinforcement learning: A survey. arXiv preprint arXiv:1609.04436.

---

> ### Author Response · Authors · 2020-11-25
> **Answer to AnonReviewer3**
>
> Thank you for the review. We submitted the revised version of the paper with an improved literature overview (Section 2) and provided a statistical derivation of the underlying method (Section 3.2). The detailed answer follows.
>
> We admit that we did not put enough emphasis on the theoretical side of TBV, hence making the impression that the method is an ad hoc heuristic rule. In fact, however, it is rather closely related to UCB and statistical hypothesis testing. Before we state how, let us make two remarks:
> The state-of-the-art planners, given a perfect model, have several mechanisms to balance exploration and exploitation, leveraging the achievements of Multi-armed Bandit theory and Reinforcement Learning (value function estimation). For instance, the implementation of MCTS used in our paper follows [13], where the in-tree exploration mechanism applies a version of upper confidence bound exploration taking the standard deviation of value ensemble predictions to measure uncertainty in estimates. This can be seen as a variant of UCB [1], UCB-V [14], or log-exp method [15]; UCT [2] being UCB applied in the tree search context). We have empirically verified that this approach performs the best among multiple choices, some of which were just mentioned. The way we train the value function ensemble follows [7], hence it can also be viewed through the Bayesian lens, similarly to Thompson Sampling. This is, however, the mechanics of the planner.
> We focus on jointly training the pair model-planner. This is an interesting task since directly trusting the planner will fail (due to model errors) and focusing on the state-space exploration to improve the model (similarly to [11] or [16]) will slow or hinder the learning of the value function. Consequently, a balance has to be struck.
>
> Coming back to TBV, we notice that the planner itself cannot distinguish between a perfect or imperfect model (at least without an appropriate mechanism). If the model is learned, it is almost impossible to avoid errors, and over-relying on the planner can lead to suboptimal actions, which then can lead to propagation of errors in the value function estimates. Having recognized that problem, we utilize a statistical hypothesis testing framework, to switch between using the planner’s exploration and a state-space exploration aiming to improve the model. The test is based on the prediction error distribution computed using the model ensemble. Such a definition is robust to the unknown scale of prediction error, as well as automates setting the threshold. Since the approach uses statistical hypothesis testing, it has a nice connection with confidence bound methods.
>
> Regarding the choice of environments, we would like to point out that TMR is a known testbed for exploration [19], and Towers of Hanoi also pose a combinatorially challenging [20]. We have demonstrated that an off-the-shelf application of planning with a learned model can fail dramatically (in the Tower of Hanoi for 7 discs, without using TBV we almost never could find a solution, see Figure 4).
>
> [13] Milos et al., Uncertainty-sensitive learning and planning with ensembles, 2019.
>
> [14] Audibert et al.,  Tuning bandit algorithms in stochastic environments, 2007.
>
> [15] Lowrey et al., Planonline, learn offline: Efficient learning and exploration via model-based control, 2019.
>
> [16] Sekar et al., Planning to explore via self-supervised world models, 2020
>
> [17] Lee et al., Sunrise:  A  simple  unified framework for ensemble learning in deep reinforcement learning, 2020.
>
> [18] Kumar et al., Discor: Corrective feedback in reinforcement learning via distribution correction, 2020.
>
> [19] Guo et al., Efficient exploration with self-imitation learning via trajectory-conditioned policy, 2019.
>
> [20] Pierrot et al., Compositional Neural Programs with Recursive Tree Search and Planning, 2019.

---

### Official Review · AnonReviewer4 · 2020-10-28
**Trust, but verify: model-based exploration in sparse reward environments**

**Rating:** 4
**Confidence:** 3

**Review:**

This paper presents a new method which can be combined with graph search algorithms to boost exploration when the uncertainty is high. This new mechanism, called TBV, can override actions given by the model to explore and verify model predictions. It is also shown in the experiments that TBV improves the model performance when combined with MCTS or BestFS. TBV utilizes graph structure of the problem and finds the solution much quicker for both MCTS and BestFS.

While the presented method is interesting with high performance, I found many editorial errors in the writing. For example, in the second paragraph of section 3.3, ‘we concentrated of exploration….’, and ‘In our experiments, such a version proved to be effective in in discrete…’, just to name a few. There are so many errors like this and the paper needs serious rewriting. Also, having a conclusion or discussion can help the structure of the paper.

Figure 3 is unclear if the blue line is without TBV with the legend ‘Right corridor visited’. It could be interesting to discuss extension of TBV into continuous environments. Reference format seems to have errors since there are underlines.

---

> ### Author Response · Authors · 2020-11-25
> **Answer to AnonReviewer4**
>
> Thank you for the review. We submitted an overhauled version of the paper, taking into account the aforementioned concerns. In particular we added the conclusion section, more references (Section 2), and described the formalism for the method (Section 3.2).
>
> Design of TBV does not rely heavily on discrete environments but we expect it would bring the most value for cases where false-loops are presented. For continuous domains such errors are likely to occur when discrete latent representation of observation is learned (for example Hafner et al 2020 found that such a latent worked the best for their model-based RL approach on Atari)
>
>
> [1] Hafner, et al. Mastering Atari with Discrete World Models, 2020

---

### Official Review · AnonReviewer2 · 2020-10-28
**Interesting and well-written paper on model-based exploration**

**Rating:** 6
**Confidence:** 3

**Review:**

This work tackled the problem of model-based RL in environments where the reward is sparse and many actions are needed to achieve some. Particularly, the authors tried to solve the issue of one-step false loop in the model, which avoids further exploration. Measuring the uncertainty about the built model through ensemble of models, they added a possibility of choosing an action different from what planner suggests, promoting exploration. The work is very-well written in general, especially sections of problem definition and related work. I also appreciate that the proposed method is compared with multiple planners and tested on two different tasks. Having said that, the main missing analysis for me is that the method was not tested on environments where false loop does not exist. Given the nature of the problem definition, i.e. learning the environment, it is counter-intuitive not to test the method on a few standard test-benchmarks without any assumption. The proposed method does not have to get the best result on environments without false loop, but it is important to see how it behaves when the built model is already good. Other than this, I have a few more questions/concerns:

1) The RANDOM parameter seems a little strange, especially because it looks too high, i.e. .5. Some analysis on different values of this (or just with and without RANDOM) on performance would be great. Also, I suspect that change in RANDOM would also change the best QR. I think a plot similar to figure 7, but for RANDOM and combination of RANDOM and QR would improve the paper.

2) The method is about one-step false loop. I would appreciate if the authors talk about multistep false loop briefly. Could it be problematic in learning? If yes, could an extension of this method work?

3) Have the authors considered a QR that changes with number of steps?

---

> ### Author Response · Authors · 2020-11-25
> **Answer to AnonReviewer2**
>
> Thank you for the review. We have submitted a new version of the paper with several improvements. In particular we added more references and described the formalism for the method (Section 3.2). The answers to your questions can be found below:
> Ad 1. We found out that the choice of threshold for RANDOM does not significantly changes the results, provided that it is 0.5 or less (see Appendix A.5).
> Ad 2. In our experiments majority (but not all) of the false transitions leading to plausible states were one-step false-loops. We expect that TBV helps with other types of errors (including multistep false-loops), since the agent replans at every step on the environment. This is indirectly confirmed by our experiments - in almost all cases agents with TBV were able to achieve a solution (in given time step limits).
> Ad 3. This is an interesting idea. For both domains on which we conducted experiments we found that the agent performance is robust to the choice of QR (see Appendix A.4), but it is likely that for other environments such a mechanism could improve the algorithm. On the downside, this introduces additional hyperparameters which may require tuning for different problems separately.

---

### Official Review · AnonReviewer1 · 2020-10-29
**A method for artificial curiosity using model uncertainty**

**Rating:** 4
**Confidence:** 4

**Review:**

The authors present a method to guide exploration that prefers to go to areas of the state space for which it is more uncertain. This uncertainty is obtained by measuring the standard deviation of the next state prediction from an ensemble of models. The authors call this the disagreement measure At each step, a search is performed and the disagreement measure is obtained for each state visited. The disagreement measure for each action is compared to the distribution for all the states visited during the search. It it is above some threshold, then the action that maximizes the disagreement measure is taken. Otherwise, it takes the action determined by the search.

The algorithm presented was unclear. What does planner.choose_action do? Is the heuristic for best-first search (BFS) the disagreement measure? I don't understand how this should help the algorithm pick a good action to take.  The paper says, "The proposed action is the first edge on the shortest path to the best node in the subgraph searched so far." How is the best node determined?

Furthermore, is this search necessary? It seems like it is mainly used as a comparison for the disagreement measure. What if the agent behaved greedily with respect to the disagreement measure all the time. Pathak et al. (2017) used a similar method, but with an inverse model.

I am not quite sure about the comparisons the authors are making. In the case of BFS search, what does it mean to do BFS search with epsilon greedy? Also, this is an artificial curiosity method where curiosity is measured by the disagreement between the ensemble of models. However, there are no comparisons to other curiosity papers, such as Pathak et al. (2017).

---

> ### Author Response · Authors · 2020-11-25
> **Answer to AnonReviewer1**
>
> Thank you for the review. We have submitted a new version of the paper with an improved presentation. In particular, we have added the pseudocode for BestFS (Appendix A.1), which should facilitate reading and make the text more self-contained. We have also expanded the related work section (Section 2), described the underlying formalism for our method (Section 3.2), and provided additional experiments (see Figure 6 in the main paper and Section A.5 in the Appendix). Below are more detailed answers:
>
> We have added pseudocodes of MCTS and BestFS to the Appendix A.1, where the  choose_action() method is included.
> The heuristic function for best-first search is indeed the disagreement measure. Before a solution to a given problem is found, there are no rewards given to the agent and the only possible strategy is to explore the state-space. The best node in the subgraph searched so far is the one which has the highest maximal disagreement (for each node we take maximal disagreement over the actions, computed the same way as STATE_SCORE in Algorithm 2). This is included in BestFS pseudocode added to this revision.
>
> We believe that model-based RL and search algorithms are important areas, which already led to spectacular results (see e.g. [1], [2]) but also have a great potential for further development.
>
> We aim to simultaneously learn the model and the planner. This requires the balance in state-space exploration (to improve the model) and planner exploration (to also improve the value function).
>
> Choosing action using epsilon-greedy equals overriding (with epsilon probability) the action proposed by the planner and replacing it with a random action (sampled uniformly).
>
> We discuss Pathak et al. 2017 and Sekar et al. 2020 in Section 2. As to the experiments we compared our work to RND which is somewhat similar to Pathak et al 2017.
>
>
> [1] Silver, David, et al. "Mastering chess and shogi by self-play with a general reinforcement learning algorithm." arXiv preprint arXiv:1712.01815 (2017).
> [2] Nagabandi et al. Deep dynamics models for learning dexterous manipulation, 2020.
> [3] Sekar, et al. Planning to explore via self-supervised world models, 2020.

---

### Decision · Program_Chairs · 2021-01-07
**Final Decision**

**Decision:**

Reject

**Comment:**

After reading the meta-reviews and the authors comment, the meta-reviewer thinks the paper is not ready for publication in a high-impact conference like ICLR. The paper is not well positioned with respect to the literature, and the proposed techniques are not well discussed in relation with predominant paradigms like optimism in the face of uncertainty.